# Control in Stochastic Environment with Delays: A Model-based Reinforcement Learning Approach

**Primary Keywords:** *Applications, Learning*

## Abstract

In this paper we are introducing a new reinforcement learning method for control problems in environments with delayed feedback. Specifically, our method employs stochastic planning, versus previous methods that used deterministic planning. This allows us to embed risk preference in the policy optimization problem. We show that this formulation can recover the optimal policy for problems with deterministic transitions. We contrast our policy with two prior methods from literature. We apply the methodology to simple tasks to understand its features. Then, we compare the performance of the methods in controlling multiple Atari games.

## Introduction

To introduce robots into our every-day life, researchers have to transfer algorithms developed in simulated environments to real environments. Existing research such as (Mahmood et al. 2018; Ramstedt and Pal 2019) has shown that applying general *Reinforcement Learning* (RL) methods to real-time control systems, such as robotic arms and self-driving cars is a challenging problem. One reason is that RL methods assume that the optimal action is directly applied to the observed state of the system. However, in real applications, the action may in fact be applied to a different state of the system. This could be due to delays in transmission, or the randomness of the system transitions. For example, in Figure 1, a real-world system is evolving while the optimal action is calculated, transmitted, etc. All these translate into *time delays* between the observed system state and the *target state*, the system state on which the action is applied. (Derman, Dalal, and Mannor 2020) show notable performance degradation of RL methods when introducing such delays in a control system. The performance further deteriorates in an environment with increased uncertainty in the next state transition.

The system evolves from the observed state to the target state by executing a sequence of prior submitted actions whose effects have not been observed yet. Previous studies (Walsh et al. 2009; Firoiu, Ju, and Tenenbaum 2018; Derman, Dalal, and Mannor 2020) use the observed state and the sequence of actions to estimate the target state and to select optimal actions based on the estimated state. These methods have shown good performance in environments with deterministic or slightly random state transitions. However, real-

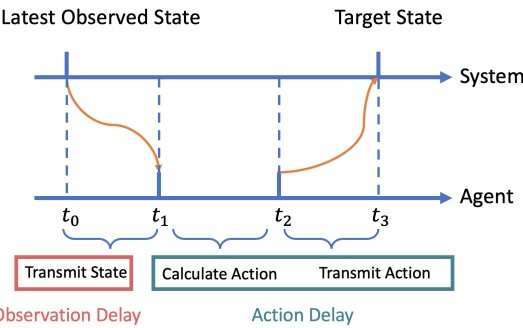

Figure 1: Illustration of control in real-time applications.

world applications can contain high level of randomness in state transitions. For example, control schemes have to cope with uncertain road/weather conditions as well as delays in transmission. In Figure 2 we illustrate a situation where we need to apply an action to the target state. Due to delays, the target state is unknown, only its probability distribution may be estimated. In this paper, we show that the aforementioned methods may not perform well in such environments where state transitions are stochastic. We design a model-based RL method to control an agent in a stochastic environment with a known constant delay. Our method learns a probabilistic model of the environment to estimate multiple possible target states and their probabilities. The method evaluates the consequences of each possible action taken.

The main contribution of our work is a new control method, Stochastic Model Based Simulation (SMBS), for problems in environments with stochastic transitions which are observed with a constant delay. We illustrate how to train the SMBS method in delayed environments. We show that the new control method recovers the optimal policy in delayed environments with deterministic transitions. In the experiments section, we illustrate the advantages of the proposed method over two baseline methods in multiple environments. We also demonstrate how the parameter of SMBS policy function can shape its risk preference.

## Preliminaries

We consider an infinite time horizon control problem with a finite action space. General RL methods model the non-

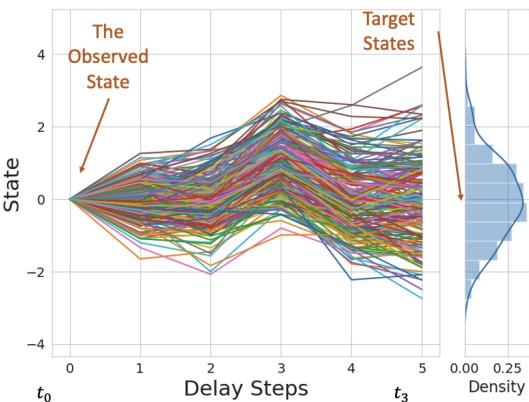

Figure 2: The stochastic environment evolution for 5 delay steps.

delayed environment as a Markov Decision Process (MDP) in (Sutton and Barto 2018). An MDP can be defined as $(\mathcal{S}, \mathcal{A}, P, r, \mu)$ where $\mathcal{S}$ is the state space, $\mathcal{A}$ is the action space, $P(S_{t+1} = s' \mid S_t = s, A_t = a)$ is the probability of transiting to next state $s'$, given current state $s$ and action $a$. The reward function $r(s, a) : \mathcal{S} \times \mathcal{A} \to \mathbb{R}$, quantifies the immediate reward obtained by applying action $a$ to current state $s$. Let $\mu$ denote the probability distribution of the initial state $s_0$. Given a deterministic Markov control policy $\pi : \mathcal{S} \to \mathcal{A}$, we can define the Q-function as the expectation of the discounted cumulative rewards for state $s$ and action $a$:

$$q^\pi(s, a) = \mathbb{E}\left[ \sum_{k=0}^{\infty} \gamma^k r(s_k, a_k) \mid s_0 = s, a_0 = a \right]$$

where $a_i = \pi(s_i)$ and $s_i \sim P(\cdot \mid s_{i-1}, a_{i-1})$ for $i = 1, 2, \ldots$. (Sutton and Barto 2018) show the optimal action value function is defined as $q^*(s, a) = \max_\pi q_\pi(s, a)$, and maximizing the function produces the optimal control policy $\pi^*(s) = \arg\max_a q^*(s, a)$.

In MDPs, actions are assumed to be applied immediately to the current states. However, in real-world applications, the action is applied to a later state than the observed state because of system delays. Previous studies (Jeong and Kim 1991; White 1988; Loch and Singh 1998; Bander and White 1999) model control problems with delays as a special case of a Partially Observable Markov Decision Process (POMDP) due to the uncertainty of the target state. A natural solution for POMDP problems is to create another MDP with an augmented state space. Following this idea, (Katsikopoulos and Engelbrecht 2003) formulate the general Augmented MDP (AMDP) for problems with constant delays in observation and action. They show that observation delays and action delays are equivalent from the perspective of the controlling agent. In our work we use *delay steps* to indicate the sum of these two types of delay. We consider a control problem with $d$ delay steps. Let $\mathcal{M}$ denote the non-delayed MDP. The corresponding augmented MDP is $\mathcal{M}_D(\mathcal{M}, d) = (\mathcal{I}, \mathcal{A}, P', r', \omega')$ where $\mathcal{I} = \mathcal{S} \times \mathcal{A}^d$, and

$r'$ is defined as $r'(I_t, a_t) = r(s_{t-d}, a_{t-d})$. Let $\tilde{q}^*$ denote the optimal Q-function for this AMDP.

Although, (Katsikopoulos and Engelbrecht 2003) show that $\tilde{q}^*$ provides the optimal control for this delayed problem, (Walsh et al. 2009) state that solving this problem in practice is difficult since the size of state space $\mathcal{I}$ grows exponentially with the number of delay steps. This issue is even more significant in modern applications, as the state and action spaces are already large, even before state augmentation.

Several recent studies use RL to find good control policies without solving the AMDP. We discuss these methods and their connections to our method in detail in the next section. We mention alternative approaches based on sim-to-real learning by (Tobin et al. 2017) and robust learning by (Pinto et al. 2017) to adapt models from simulations to real world robots. Our work focuses on solving delay problems using RL. The methodology we develop could complement a sim-to-real transfer but combining the ideas is beyond the scope of this work.

## Stochastic Model Based Simulation (SMBS)

The methods developed by (Derman, Dalal, and Mannor 2020; Walsh et al. 2009; Firoiu, Ju, and Tenenbaum 2018) can be classified as instances of deterministic planning. A deterministic planning strategy consists of two components: a model of the system dynamics, and a non-delayed policy function. This strategy obtains a single estimate of the target state. Then, based on this estimate, the non-delayed policy function selects an optimal action to be applied to the target state. (Walsh et al. 2009) state that this approach performs well in deterministic tasks. Note that a fundamental requirement is the single estimate of the target state. In stochastic environments, the estimator of the target state has a distribution which may have a large variance. As the action is chosen based on the estimator, this may lead to sub-optimal actions if the estimator is far from the actual state. Therefore, we must consider the effect of the action taken on multiple possible target states, rather than a single one.

To expand the deterministic planning method for stochastic environments, we develop a stochastic model based simulation (SMBS) method. This method develops a strategy which consists of a probabilistic model of the system and a value-based non-delayed policy function. This policy function is generated by an optimal Q-function. We denote this non-delayed optimal Q-function with $q^*$. The model of the system, denoted as $\mu(s, a)$, maps a given state/action pair to a probability measure on the state space $\mathcal{S}$. Suppose the environment has $d$ delay steps. We form an augmented state using the delayed observation and a sequence of actions, $I_t = (s_{t-d}, a_{t-d}, \cdots, a_{t-1})$. Note that $s_{t-d}$ is the latest observation of the system state, and the effect of $\{a_{t-d}, \cdots, a_{t-1}\}$ has not been observed yet. As shown in Figure 3, we sample $M$ estimates of the target states, $\{s_t^{(i)}\}_{i=1,\cdots,M}$ using the one-step transition model $\mu(s' \mid s, a)$. To be specific, for each path $i$, we let the initial state $s_{t-d}^{(i)} = s_{t-d}$. Then, we recursively sample the estimates of the next states $s_{t-d+k+1}^{(i)}$

using $\mu(\cdot \mid s_{t-d+k}^{(i)}, a_{t-d+k})$ for $k = 0, 1, \cdots, d-1$. Finally, we select the action using the following policy function

$$a_t = \pi_1(I_t) = \arg\max_{a \in \mathcal{A}} \left( \bar{Q}_M(a) - \alpha \hat{Q}_M(a) \right),$$

$$\text{where } \bar{Q}_M(a) = \frac{1}{M} \sum_{i=1}^{M} q^*(s_t^{(i)}, a), \qquad (1)$$

$$\hat{Q}_M(a) = \sqrt{\frac{1}{M-1} \sum_{i=1}^{M} (q^*(s_t^{(i)}, a) - \bar{Q}_M(a))^2}.$$

The function $\bar{Q}_M(a)$ estimates the average state action value for action $a$. The risk of executing action $a$ when the target state is unknown is measured using $\hat{Q}_M(a)$. Mathematically, $\hat{Q}_M(a)$ is the sample standard deviation of the Q-value of the target state. The hyper-parameter $\alpha$ controls the importance of $\hat{Q}_M(a)$.

The policy function of SMBS (equation 1) uses the mean and standard deviation of the sampled Q-values. Intuitively, it will select optimal actions by maximizing the expected action value while minimizing the deviation of the Q-values from their mean value. Algorithm 1 introduces the pseudo-code used to implement the SMBS method.

---

**Algorithm 1: Stochastic Model Based Simulation Policy**

1: **Input**: A trained Q-function $q^*$, a trained system model $\mu$, $I_t = (s_{t-d}, a_{t-d}, \cdots, a_{t-1})$, $\mathcal{A}$.
2: **Output**: the action $a_t$ which is applied on state $s_t$.
3: Initialize: a state container $\mathcal{D}$, an expected Q value list $\mathcal{V}$
4: Planning:
5: **for** $i = 1, 2, \cdots, M$ **do**
6:     Let $s_{t-d}^{(i)} = s_{t-d}$;
7:     **for** $j = 0 : d - 1$ **do**
8:        $s_{t-d+j+1}^{(i)} \sim \mu(\cdot|s_{t-d+j}, a_{t-d+j})$;
9:     **end for**
10:     $\mathcal{D} \leftarrow s_t^{(i)}$;
11: **end for**
12: Evaluating:
13: **for all** $a$ in $\mathcal{A}$ **do**
14:     Initialize a new list $\mathcal{L}$ for all state-action values for action $a$;
15:     **for all** $s_t^{(i)}$ in $\mathcal{D}$ **do**
16:        $\mathcal{L} \leftarrow q^*(s_t^{(i)}, a)$;
17:     **end for**
18:     $\mathcal{V}[a] = mean(\mathcal{L}) - \alpha \cdot std(\mathcal{L})$;
19: **end for**
20: **return** $\arg\max_a \mathcal{V}[a]$.

---

We present a training procedure for the SMBS method in delayed environments in Algorithm 2. In the sample collection step, the delayed action $a_{t+d}$ is selected using the SMBS policy in 1. We record the transition sequence of the system and determine the actions corresponding to respective state transitions. This will produce the non-delayed

---

**Algorithm 2: Training the Q-function $q$ and the system model $\mu$ for the SMBS method**

1: **Input**: An interactive environment $E$.
2: **Output**: a Q-function $q$ and a model of the system dynamics $\mu$.
3: Initialization: randomly initialize the parameter in $q$ and $\mu$, a data container $C$ for transition samples.
4: /* Collecting Samples */
5: $I_d = (s_0, a_0, \cdots, a_{d-1}) \leftarrow E$
6: **for** $t = d, d + 1, \cdots, N$ **do**
7:     $a_t \leftarrow \text{SMBS}(I_t; q, \mu)$;
8:     $s_{t-d+1}, r_{t-d} \leftarrow E(a_t)$;
9:     $C \leftarrow (s_{t-d}, a_{t-d}, s_{t-d+1}, r_{t-d})$; /*non-delayed transitions*/
10:     $I_{t+1} \leftarrow (s_{t-d+1}, a_{t-d+1}, \cdots, a_t)$;
11: **end for**
12: /* Training */
13: **for all** sample $c$ in $C$ **do**
14:     Update $q$ using DDQN in (Van Hasselt, Guez, and Silver 2016);
15:     Update $\mu$ using Maximum Likelihood Method;
16: **end for**
17: **return** $q, \mu$

---

transitions $(s_t, a_t, s_{t+1}, r_t)$ which are used to train $q^*$ and $\mu$. Similar procedures are also used by (Schuitema et al. 2010; Derman, Dalal, and Mannor 2020).

The main difference SMBS and the planning policy in (Derman, Dalal, and Mannor 2020) is that the policy of (Derman, Dalal, and Mannor 2020) plans for the most likely next state. As a result, once trained the planned action path is always the same for the same input $I_t$. This is why we term this planning method as deterministic. In contrast, SMBS (equation (1)) obtains multiple target state estimates by sampling trajectories using a probabilistic model of the system. The action is selected to maximize the *average* Q-value subject to the penalty term.

Even though the SMBS method is designed to accommodate a stochastic environment, it also works in deterministic environments. In fact, the next result establishes that the SMBS method provides the optimal control. A proof of this theorem is provided in the supplementary material.

**Theorem 1.** *Assume a discrete-time MDP with an infinite time horizon. The Markovian movement is deterministic, i.e., for arbitrary $(s, a) \in \mathcal{S} \times \mathcal{A}$, $t \geq 0$, there exists an $s' \in \mathcal{S}$ such that $P(S_{t+1} = s' \mid S_t = s, A_t = a) = 1$ for all $t = 0, 1, \ldots$ Then, the policy function of the SMBS method (1) is equivalent to the following optimal policy:*

$$\pi_{opt}(I_t) = \arg\max_{a \in \mathcal{A}} \tilde{q}^*(I_t, a), \qquad (2)$$

*where $\tilde{q}^*$ denotes the optimal Q-function for the AMDP.*

When parameter $\alpha$ is set to 0, the SMBS method is reduced to a Monte-Carlo procedure that estimates $\mathbb{E}_s[q^*(s_t, a) \mid I_t]$. This conditional expectation is associated with the probability measure for $s_t$, given $I_t$. Thus, the policy (1) in the SMBS method approximates the following

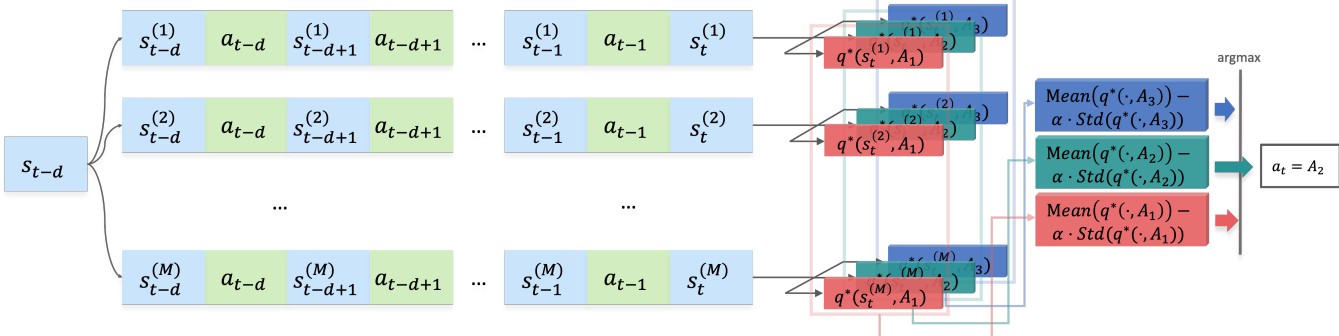

Figure 3: An illustration of the policy function of the SMBS method.

policy function

$$a_t = \arg\max_{a \in \mathcal{A}} \mathbb{E}_s[q^*(s_t, a) \mid I_t].  \quad (3)$$

When the problem has a small discrete state space, the model of the system may be expressed using one-step transition matrices. In such cases, we compute the distribution of the target state using these transition matrices. Then, we calculate the expectations in (3) for all actions $a \in A$ and select the action with the highest expected value. We note that this procedure has been analyzed by (Agarwal and Aggarwal 2021). Interestingly, although the two policy functions are similar, the two methods have been developed independently from different perspectives. When the problem has a large discrete state space or a continuous state space, the SMBS method models the system model using an artificial neural network (ANN). Thus, instead of learning transition matrices, the algorithm focuses on learning the parameters of the ANNs approximating the model. When the problem solved involves a complicated state space, this change avoids multiplications between large matrices which are computationally expensive and inefficient. Our method can solve complex tasks by incorporating parametrized Q-functions which are trained using well-known deep RL methods such as DQN in (Mnih et al. 2015), DDQN in (Van Hasselt, Guez, and Silver 2016), Dueling DQN in (Wang et al. 2016), etc.

We also provide a probabilistic bound for the SMBS policy in (1) when $\alpha = 0$.

**Theorem 2.** *Assume a discrete-time MDP with a positive reward function and a finite discrete action space $\mathcal{A}$. For any $a \in \mathcal{A}$ and augmented state $I_t \in \mathcal{I}$, assume the random variable $q^*(s_t, a)$ has mean $\bar{Q}(a)$ and variance $\hat{Q}(a)^2$. Then, for $\delta > 0$, we have*

$$P\left( \max_{a \in \mathcal{A}} \bar{Q}_M(a) \le \frac{1}{|\mathcal{A}|} \mathbb{E}\left[ V^*(s) \mid I_t \right] \right.$$
$$\left. - \frac{\delta}{\sqrt{M}} \max_{a \in \mathcal{A}} \hat{Q}(a) \right) \le \frac{|\mathcal{A}|}{\delta^2}$$

The proof of this theorem is provided in the supplementary material.

If the variance $\hat{Q}(a)^2$ exists for any $a \in \mathcal{A}$, its sample estimates converges to 0 when the sample size $M$ goes to infinite. This implies that if we plan for sufficiently large amount of paths, the value of SMBS policy function is not worse than $\frac{1}{|\mathcal{A}|}\mathbb{E}\left[ V^*(s) \mid I_t \right]$ with high probability.

## Experiments

In order to understand how the SMBS method performs in delayed environment with different levels of randomness in transitions. We train and test the SMBS method with other baseline methods in multiple tasks. We choose the following baseline methods for comparison.

1. AMDP in (Katsikopoulos and Engelbrecht 2003), which forms an AMDP and directly solve the problem using Double DQN by (Van Hasselt, Guez, and Silver 2016).

2. Delayed-Q in (Derman, Dalal, and Mannor 2020), which is, to the best of our knowledge, the latest deterministic planning method. This method follows the following policy

$$\pi_2(I_t) = \arg\max_{a \in \mathcal{A}} q^*(\hat{s}_t, a),  \quad (4)$$

where $\hat{s}_t$ is obtained by a recursive propagation using a deterministic model of the system $\hat{m}(s, a) = \arg\max_{s'} P(S_{t+1} = s' \mid S_t = s, A_t = a)$. That is,

$$\hat{s}_{t-d+1} = \hat{m}(s_{t-d}, a_{t-d}),$$
$$\hat{s}_{t-d+2} = \hat{m}(\hat{s}_{t-d+1}, a_{t-d+1}),$$
$$\cdots$$
$$\hat{s}_t = \hat{m}(\hat{s}_{t-1}, a_{t-1}).$$

We note that while the Delayed-Aware Trajectory Sampling (DATS) method in (Chen et al. 2020) may be used as a baseline comparison, it does require knowledge of the exact reward function. As the other three methods learn the reward function, we excluded DATS from our analysis.

When comparing the SMBS with the baseline methods, we are trying to address the following questions:

1. Does the SMBS method obtain better average rewards?

2. Is there less performance degradation of the SMBS method than the baseline methods when the number of delay steps increases and when randomness in the environment increases?

## Tasks

We perform the experiments on these four tasks: Stormy and Swampy Road, Frozen Lake (4-by-4), Cartpole, and Puddle World.

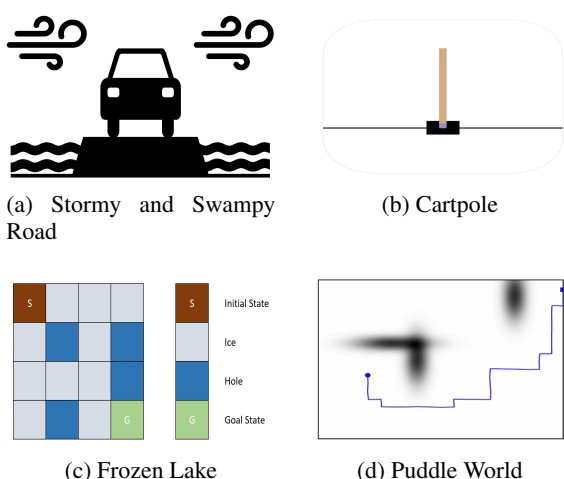

(a) Stormy and Swampy Road

(b) Cartpole

(c) Frozen Lake

(d) Puddle World

Figure 4: Illustrations of tasks used for comparison.

*Stormy and Swampy Road* is a simple control problem with a 1-dimensional state space and a discrete action space with 4 actions. Figure 4a illustrates this environment. The task requires the agent to maneuver a car on a narrow road during a storm. The road is through a large swamp. The car can fall off the road and get stuck in the swamp on both sides of the road. There are four actions possible representing 4 forces with different directions and magnitudes, i.e., steering to left/right aggressively/mildly. If the car runs into the swamp, an aggressive steering to the road has a higher chance to move the car back to the road than a mild steering. If the agent cannot return the car to the road immediately, the agent will receive a large penalty. The storm randomly pushes the car to the left or the right. The level of randomness is adjusted by a parameter $r$. A higher $r$ increases variability of transition. We test $r = 0.05/0.1/0.15$ in our experiments.

*Frozen Lake* is a maze-like problem which is implemented in (Brockman et al. 2016). Figure 4c illustrates the 4-by-4 Frozen Lake task. The goal is to reach the Goal State (G) starting from the Initial State (S). The agent goes back to the Initial State if at any time reaches a Hole state (in blue in Figure 4c). This environment has a slippery parameter $p$ ($1/3 \leq p \leq 1$) which controls the randomness of the transition. This parameter indicates the probability of moving to the intended target square. That is, if the agent chooses to go right, it will arrive to the state to its right with probability $p$. Otherwise, the agent may move to the two other adjacent states (up and down) with equal probabilities $\frac{1-p}{2}$. We denote $r = \frac{1-p}{2}$, we test $r = 0.05/0.1/0.15$ in our experiments.

*Cartpole* is a classic benchmark task, and has been implemented in (Brockman et al. 2016). The agent applies left-/right forces onto the cart to keep the pendulum balance. A normally distributed noise with zero mean is added to the force. The standard deviation of the noises is a parameter $r = 0.1/0.2/0.3$.

*Puddle World* is another classical task mentioned in (Degris, White, and Sutton 2012). The goal is to navigate to the goal state (1.0, 1.0) on a 2D world at the soonest without stepping in the high penalty zone (grey area in Figure 4d). The movement noise is controlled by a parameter $r = 0.005/0.01/0.02$.

## Training/Evaluation Procedure

For each environment setting (level of randomness and number of delay steps), we train 5 models (with different random seeds attached to the clock) using each of the three methods. Each model is trained in the delayed environment with $10^5$ steps. The policy with the highest average reward is recorded for evaluation. Each model is evaluated using the $10^4$ steps. We report the average rewards by the top four models in the next section.

The system dynamics model and the Q-function are trained using the dataset collected from the delayed environments. For the Stormy and Swampy Road and for the Frozen Lake environments we estimate the transition matrices using the observed frequency of transition. For the Puddle World and for the Cartpole environments we model system dynamics using Gaussian based probabilistic neural networks, which are often used for continuous control problems in (Duan et al. 2016; Mnih et al. 2016). The neural networks have two layers, each layer has 64 units. For the SMBS method, we plan 50 trajectories for each decision ($M = 50$). The small number of paths is chosen so that the algorithm converges in a reasonable amount of time. The $\alpha$ parameter controls the importance of the variability of the estimated future cumulative rewards. A large alpha produce policies that produce stable future rewards. However, these may not be optimal from the perspective of maximizing reward. After an extensive number of experiments, we choose $\alpha = 0.01$ in our studies.

## Results

The average rewards from three methods are reported in Figure 5. Figure 5a shows SMBS and AMDP both perform better than Delayed-Q in all settings. SMBS outperforms AMDP when the environment has 20 steps of delay and the random factor $r = 0.15$. This result is not surprising as the task is simple and easy to solve. AMDP can produce the optimal strategy given sufficient amount of samples ($10^5$ steps). In Figure 5b, the bar plots indicate that SMBS has the overall best performance among three methods. The performance degradation is relatively smaller than the other methods. When the level of randomness is small ($r = 0.1/0.15$), AMDP and SMBS have better performance than Delayed-Q. When $r = 0.2$, the performance of AMDP significantly degrades, SMBS still outperforms the other two methods. Figure 5c and 5d both indicate that SMBS and Delayed-Q have similar performance and are consistently better than AMDP.

Moreover, to understand how the SMBS method can outperform the Delayed-Q method in Stormy and Swampy

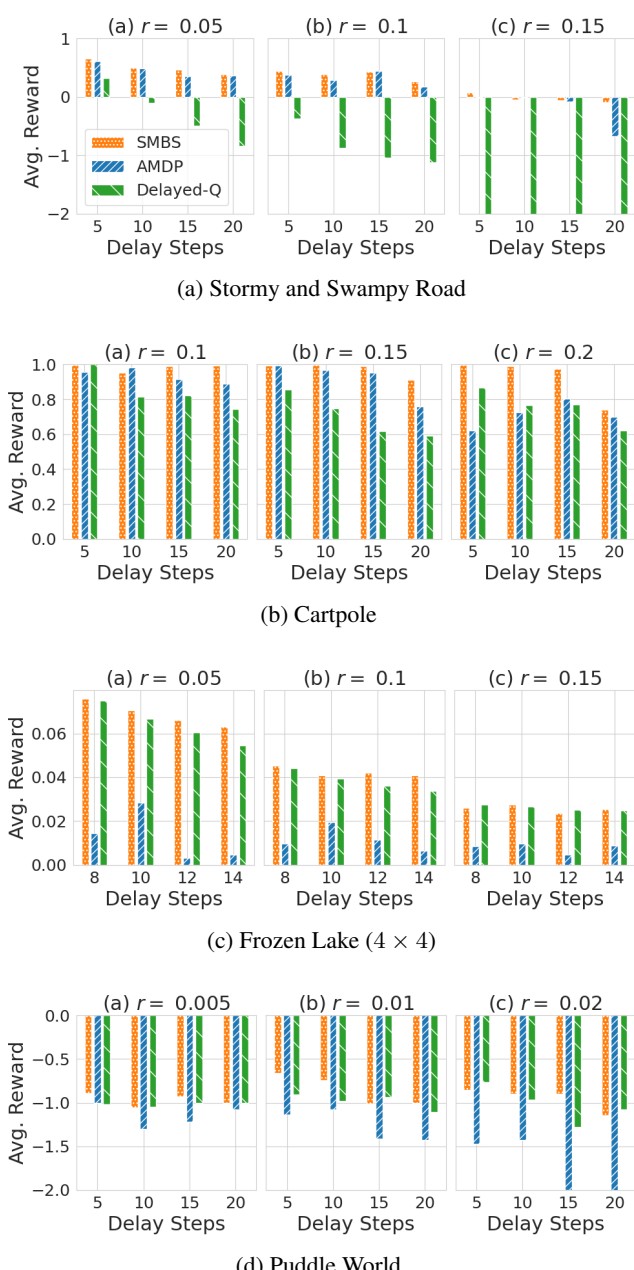

Figure 5: Illustrations of tasks used for comparison.

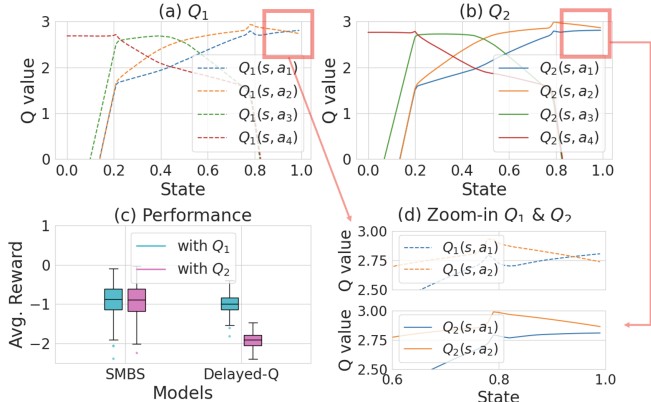

Figure 6: An illustration of robustness of the SMBS method with respect to the Q-function approximation errors. Figures (a) and (b) show the values of two Q-function approximations which have slight discrepancies magnified in (d). Figure (c) shows the difference in performance for the two methods.

different actions when the estimated target state is close to 1. In contrast, the SMBS policy is less affected by small changes in the Q-function since it considers the expected Q-value.

## Atari Learning Environments

Atari Learning Environments (ALE) in (Bellemare et al. 2013) offers intricate environments for training and testing RL methods. (Mnih et al. 2015) has shown that RL agents can surpass human-level strategies using RL methods. With delayed observations, (Derman, Dalal, and Mannor 2020) shows the Delayed-Q method can outperform baseline methods such as AMDP and Obvious-Q method on ALE.

We apply our method to ALE and compare it with the AMDP method and the Delayed-Q method. Across all experiments, additional randomness in movements is introduced by setting a 0.2 probability for *sticky actions*. Sticky actions simulate scenarios where the controller ignores the input action and repeats the prior action. We perform training and testing of the three methods across 7 Atari games when the number of delay steps equals to 5 and to 25. In each experiment setting, we train models with the three methods with the same amount of steps, and we load the best-performing model for evaluation. In the SMBS method, we set the number of planning paths $M$ as 20 to create a trade off between efficiency and accuracy. We set the risk preference parameter $\alpha$ as 0 because we would like to only maximize the expected reward.

The evaluation results are reported in Figure 7. It can be seen that the AMDP method performs worse than the other two methods in most of the games. The AMDP method also suffers from stronger performance degradation than the two other methods when the number of delay steps increases from 5 to 25. This is evident in Freeway and RoadRunner. The performance of Delayed-Q and SMBS is consistent when the number of delay steps increases. Their two

Road task, we evaluate the both methods using the same system model and the same Q-function. Figure 6 shows a comparison of performance between SMBS and the Delayed-Q with a slight change of the estimated Q-function. The two Q-functions in Figure 6a and 6b are two non-optimal Q-functions that are consecutively recorded during training ($Q_1$ is closer to the real Q-function than $Q_2$). Figure 6d displays the main difference between two Q-functions. When the state is close to 1, $Q_1$ indicates the action $a_1$ has a higher value than $a_2$, while $Q_2$ indicates $a_2$ has higher value than $a_1$. As a result, the policy of Delayed-Q method produces

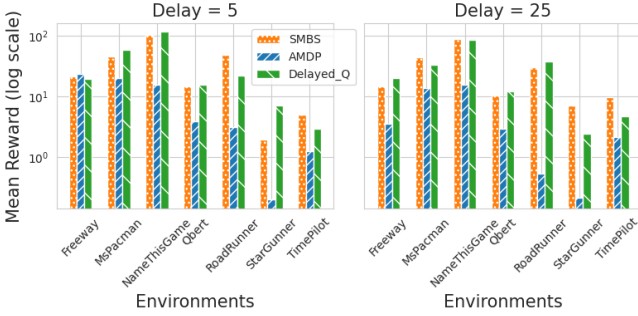

Figure 7: Comparisons of SMBS, AMDP, and Delayed-Q in different Atari games with delayed feedback.

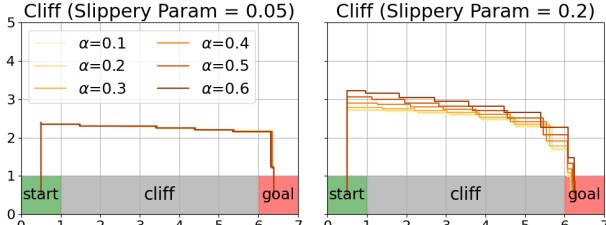

Figure 8: Average paths in Cliff environments with different risk preference parameters ($\alpha$). Left: Paths overlap in a more deterministic environment, indicating minimal influence of $\alpha$. Right: Divergent paths in a more stochastic setting; higher $\alpha$ values result in paths farther from the cliff edge, depicted by darker colors.

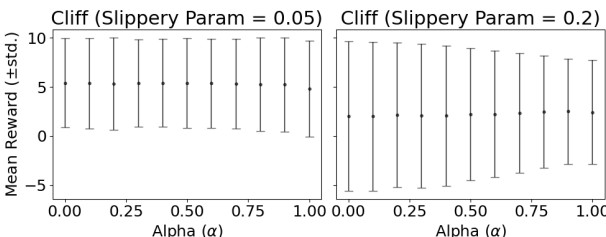

Figure 9: Comparison of the expected SMBS policy reward for varied risk preference parameters (0 to 1) in the Cliff environment. Left plot: Slippery parameter = 0.05; Right plot: Slippery parameter = 0.2.

algorithms performance is comparable in Freeway, MsPacman, NameThisGame, and Qbert. When the number of delay steps equals 5, Delayed-Q performs better than SMBS in StarGunner, and SMBS performs better in RoadRunner and TimePilot. When the number of delay steps increases to 25, SMBS outperforms Delayed-Q in StarGunner, and TimePilot.

### Risk Parameter $\alpha$

To explore how different risk preferences impact the policy function in 1, we examine the policy function in *Cliff*, a classic control problem from (Sutton and Barto 2018). In Cliff, the agent navigates a maze-like path from start to end (refer Figure 8). The agent needs to avoid falling off the cliff during the movement. When the agent moves, the agent has a chance to slip towards the cliff. Due to delays in observations in this problem, the actual location of the agent is unclear when the action is chosen. Therefore, the agent faces the choice between a risky move to the right or a safer move upwards. We select this task to demonstrate the impact of the risk preference parameter because the risk involved in this problem is easy to understand. A more risk-averse agent would prefer moving away from the cliff due to the risk of falling.

We construct the SMBS policy functions using the optimal Q-function and different risk preference parameters $\alpha$. We then run these policy functions in the Cliff environment. Figure 8 presents the average path for different risk preference parameters. When the slippery parameter is small, the risk preference parameter has small impact on the policy functions and the resulting paths are very similar. However, when the slippery parameter is large, thus more randomness in movements, it can be seen that higher risk preference parameters lead to paths distant from the cliff edge. This demonstrates that policies with higher $\alpha$ values exhibit greater risk aversion. In Figure 9 we illustrate the SMBS method performance for varying $\alpha$ values in the Cliff environment. In a more deterministic setting (left graph with slippery chance 5%), we see little difference in the policy function performance regarless of the risk averse parameter $\alpha$. In Figure 9 right increasing alpha produces less variable results which means that the policy becomes more and more conservative. This aligns with expectations, as a risk-averse agent tends to move farther from the cliff to reduce the risk of falling, consequently prolonging the time taken to reach the goal.

## Conclusion

In this study, we investigate control problems in stochastic environments with constant delayed feedback. We develop a new method (SMBS) designed to approximate optimal control for such problems. We show that the SMBS method is optial when the system has deterministic movement. We compare the performance of the SMBS method, with two other baseline methods using 4 classical control environments and 7 Atari Learning Environments. We observe performance degradation as the number of delay steps increases and as the level of randomness in transitions is increased. Our experiments show that the SMBS method outperforms AMDP in most experiments and is no less than the Delayed Q method in most Atari games. Further, the SMBS method is more robust to errors in estimation of the Q-function. We also showcase the impact of the risk preference parameter in the SMBS policy function. This parameter may be used to further tune the agent behavior in response to perceived delays.

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
