# OpenReview forum: "Control in Stochastic Environment with Delays: A Model-based Reinforcement Learning Approach"
_icaps-conference.org/ICAPS/2024/Conference — ICAPS 2024_

### Official Review · Reviewer_qEDa · 2024-01-16

**Significance And Importance:** 2
**Soundness:** 3
**Novelty:** 3
**Clarity:** 3
**Overall Evaluation:** 2
**Confidence:** 3

**Weaknesses:**

1: Minor weaknesses that are easily fixable.

**Contributions Of The Paper:**

The paper introduces an approach for decision-making in stochastic environments, particularly focusing on uncertain scenarios and delayed feedback. This method, termed Stochastic Model-Based Simulation (SMBS), is an interesting approach. It employs a probabilistic framework that anticipates a spectrum of possible future states rather than relying on single-point estimates. This approach allows for more robust and flexible decision-making in unpredictable environments.

The central idea of SMBS is integrating a probabilistic model with a value-based policy function, which is optimized using reinforcement learning techniques. This integration enables the system to account for the uncertainty and variability of real-world environments.
The methodology employs a probabilistic system model (μ(s, a)) and an optimized value-based policy function (q*). SMBS handles uncertainty and delays by forming an augmented state from delayed observations and past actions, and it addresses the stochastic nature by sampling multiple future state estimates using the transition model μ(s | s, a).

In action selection, SMBS balances the mean estimated value of actions (QˉM(a)*Q*ˉ*M*(*a*)) and the associated risk or uncertainty (Q^M(a)*Q*^*M*(*a*)), adjusted by a risk parameter (\alpha). This policy function maximizes the expected reward while considering the variability of outcomes, differentiating it from deterministic strategies.

For training in complex environments, SMBS employs deep reinforcement learning methods and artificial neural networks, optimizing the computational process. The policy's reliability is supported by a probabilistic bound (Theorem 2), ensuring that its performance aligns closely with the expected value function, given a sufficient number of paths.

The authors conduct a series of experiments using various tasks such as cartpole, frozen lake, swampy road, and puddle world domains, including challenging Atari games, to test the efficacy of the SMBS framework. These experiments are designed to assess the model's performance in environments with different levels of randomness and delay in feedback. The results demonstrate the superiority of SMBS over traditional methods, especially in scenarios where conventional models struggle due to uncertainty and delayed responses.

Furthermore, the paper delves into analyzing a risk parameter (\alpha) within the policy function. This exploration is critical as it sheds light on how different risk considerations can impact decision-making strategies in stochastic environments. The authors illustrate how tweaking this parameter can lead to different policy behaviors, thereby enabling the management of risk in uncertain conditions.

**Ethical Considerations:**

(1) Not Applicable: The paper does not have any ethical considerations to address

**Nomination For Best Paper:**

No

**Questions For Authors:**

1. Please comment on the time and space complexity of the proposed method and how well the method will scale for real-world robotic tasks such as manipulation and long-horizon problems.

2. Please comment on the papers I mentioned in the weakness section and explain how they differ from the proposed approach.

**Reproducibility:**

4: Authors promise to release code and domains (whichever apply).

**Strengths Of The Paper:**

1. The development of the Stochastic Model-Based Simulation (SMBS) for control in uncertain environments with delayed feedback is a novel contribution. It addresses a gap in current methodologies.

2. The paper demonstrates a strong mathematical foundation in its formulation of the SMBS, particularly in integrating probabilistic modeling with a value-based policy function optimized through reinforcement learning. The appendix also included mathematical proofs of the theorems presented in the paper. The method seems interesting, and the illustration and the mathematical formulation seem correct and well-elucidated.

3. The experimental setup, using various tasks, including Atari games, is well-designed. It effectively demonstrates the model's robustness in different stochastic scenarios with delayed feedback.

4. The model's ability to handle stochastic environments suggests its applicability in real-world scenarios, which is an important aspect, specifically for non-stationary and stochastic environments.

**Weaknesses Of The Paper:**

1. Complexity in Real-World Applications: The method shows promise in simpler grid-world simulations, but its scalability and computational feasibility in complex real-world tasks, such as robotic manipulation with vast state and action spaces, are not addressed. The proposed model's approach of selecting actions based on multiple trajectories from the current state might face tractability issues in long-horizon tasks, such as those in robotic manipulation. These tasks often involve a vast number of potential future states, making the computation of multiple trajectories computationally intensive. In such scenarios, the model's feasibility and efficiency could be significantly challenged. Could the authors provide a complexity analysis or comment on the potential computational limitations in more intricate scenarios?

2. Assumption of Constant Delay: The paper assumes a fixed delay, which simplifies the model but may not accurately reflect the variability found in real-world delays. Can the authors discuss the implications of this assumption and potential adaptations for variable delays?

3. Comparative Analysis with Related Work: The paper lacks a thorough comparison with relevant research in lookahead learning with delay, specifically papers by Han et al. (2022) and Lancewicki et al. (2023). How does the proposed method compare with these approaches, and why might these methods not be adaptable to the authors' framework?

Han, B., Ren, Z., Wu, Z., Zhou, Y. & Peng, J.. (2022). Off-Policy Reinforcement Learning with Delayed Rewards. Proceedings of the 39th International Conference on Machine Learning, in Proceedings of Machine Learning Research 162:8280-8303 Available from https://proceedings.mlr.press/v162/han22e.html.

Lancewicki, T., Rosenberg, A., & Sotnikov, D. (2023). Delay-Adapted Policy Optimization and Improved Regret for Adversarial MDP with Delayed Bandit Feedback. *ArXiv*. /abs/2305.07911 [Published at ICML 2023]

Rosenberg, A., Hallak, A., Mannor, S., Chechik, G., & Dalal, G. (2023, June). Planning and learning with adaptive lookahead. In Proceedings of the AAAI Conference on Artificial Intelligence (Vol. 37, No. 8, pp. 9606-9613).

---

> ### Author Rebuttal · Authors · 2024-01-27
>
> Thank you very much for the review as well as for the references provided.
>
> Our method should solve the problem in polynomial time as the level of complexity is similar to work it extends [Walsh et al. 2009] and [Derman et al. 2021]. The extra complexity comes from the additional planning procedure. However, the paths from the planning procedure are independent from each other and they can be easily parallelized.
>
> Transferring algorithms from simple experiments to real life applications is always challenging. In this work we transfer the algorithm to difficult Atari games. While quite not as involved as real robotics applications, these games require working with a continuous state spaces. As pointed out, the weakness of the methodology is in the fixed number of delay steps, as for robotics applications we have variable delay. In fact, this is how the problem originated. We had to control a jumper by sending actions through the wireless network and our university has a pretty unstable connection. Trying to solve this problem we first reduced the problem to fixed delay: the expected number of delay steps. This is the current paper and the algorithm we created. We plan to expand the work to stochastic delays in future work.
>
> Thank you very much for the references provided. [Han et al. 2022] investigate control problems with delayed reward. However, the transition process is still Markovian and fully observable. In contrast, in our work we solve problems with delayed observations and actions. Despite the fact that the reward is also delayed for us, the methodology in this paper is not applicable to our problem.
> Similarly, the problem studied in [Lancewicki 2023] is different. This paper studies delayed feedback including policy updates and exploration without observing feedback. The problem is motivated by recommender systems. In contrast, in our study despite the delay we have to observe feedback (reward) as this is how the policy learns. [Rosenberg 2023] propose algorithms which perform adaptive selection of planning horizon in a learning framework. This work is really interesting and we thank the reviewer for pointing it to us. We believe ideas from this work may be incorporated in our future work. Although the method is not directly applicable to our problem as we deal with a fixed number of delay steps, the adaptive planning which dynamically changes the number of planning paths and depth can potentially help reduce the complexity of our method.

---

### Official Review · Reviewer_MG1q · 2024-01-21

**Significance And Importance:** 2
**Soundness:** 3
**Novelty:** 2
**Clarity:** 3
**Overall Evaluation:** 1
**Confidence:** 4

**Weaknesses:**

1: Minor weaknesses that are easily fixable.

**Contributions Of The Paper:**

This paper presents a new approach called Stochastic Model-based Simulation Policy (SMBS) that handles stochasticity in domains with delayed feedback (the delay steps are fixed). Building upon augmented MDPs (AMDPs) the authors propose the use of model-learning and planning techniques that are aware of the stochasticity in the domain. Prior work only addresses planning in deterministic settings where the determinism comes from single-outcome determinization.

In order to handle stochasticity, SMBS uses a hyper-parameter M to compute M different simulated trajectories.  The q-values of the target states of these M different trajectories are then averaged out the compute the policy. The authors also propose a risk-aware component in the form of the std. deviation of the q-values of the M different target states.

The authors propose a training procedure for learning both the q-values and the model which are then used in SMBS. A theoretical analysis is presented followed by an empirical evaluation on their approach on 4 different domains a a variety of atari games.

**Ethical Considerations:**

(5) Excellent: The paper comprehensively addresses all of the applicable ethical considerations

**Nomination For Best Paper:**

No

**Questions For Authors:**

I think that the work is interesting but the experiment design in the paper might be poor and might not be well-suited to showcase the strengths of SMBS well. One immediate improvement would be to use higher values of r to induce even more randomness.

Q1. Could you please comment on W2. Is there any intuition as to why the performance on Frozen Lake and Puddle world are quite close. For example, when r=0.05, as the delay steps increase SMBS starts outperforming Delayed-Q but surprisingly for r=0.1 and r=0.15 it is not so prominent. One would expect delayed-Q to start performing worse as d increases. An ablation of sensitivity to M is not provided and it would be useful to have such info. Similarly, please also provide an intuitition on why SMBS worsens in RoadRunner while Delayed-Q performance increases?

General comment here: I think Fig 8 and Fig 9 could be left to supplement and you could use the space to provide a better analysis on why these performance differences occur particularly as "d" increases since one would expect SMBS to shine in such cases with high values of "r".

Q2. Could you please comment on W3?

Q3. In line 8 of Alg. 1, why are you sampling from \mu? Since you have already computed the probability distribution you can always grow a tree of depth d making the approach independent of M as well. Is there a particular reason for sampling?

Post Rebuttal
===========
Thank you for clarifying my questions. I have raised my score to a weak-accept. If this paper were to get rejected, I'd encourage the authors to tighten the empirical evaluation that showcase the performance of SMBS over Delayed-Q better and resubmit. (Handling of continous environments is a strength and I hope that the authors are able to emphasize that as well). I wish the authors all the best.

**Reproducibility:**

5: Code and domains (whichever apply) are already publicly available

**Strengths Of The Paper:**

1. The paper is well-written and is quite clear
2. The ideas presented are convincing and the handling of stochasticity makes it widely applicable
3. I liked the risk-averse component that can guide towards safer and/or stable policies
4. Fig. 6 showcases some of the benefits of their approach over deterministic methods and provides a convincing argument in such a case.

**Weaknesses Of The Paper:**

Unfortunately, while the work is quite interesting I think there are some weaknesses that might need to be fixed.

W1. This is a minor comment but using \citet for in-line text citations would be better for listing ciations.

W2. It seems from the empirical evaluation, that this approach is not particularly well-suited to heavily outperform the closest baseline: Delayed-Q which also utilizes planning via single-outcome determinization.

For example, even as the total noise and delay steps increase, the overall performance in domains such as Frozen Lake and Puddle World and many of the Atari games remains quite similar and does not best showcase the merits of SMBS. Similarly, for RoadRunner in Atari, performance deteriorates for SMBS as "d" increases.

W3. Fig. 9 (right) is unclear. One would expect that as \alpha increases, the avg. reward goes significantly down but the mean rewards remain relatively the same even if the std. deviation decreases slightly.

---

> ### Author Rebuttal · Authors · 2024-01-27
>
> Thank you very much for your comments. We found all questions well thought out. Addressing them will improve the paper substantially. We will make changes as specified below in the final submission.
>
> 1. The "citet" option - eternally grateful for making us aware of this natbib option.
>
> 2. When r increases, as the state space is quite small, introducing randomness makes the problem very hard. Frozen lake has 16 total states and transitions are only to neighbouring states. Both models' performance suffers. We recreated the frozen lake experiment for a 8x8 grid. The difference between SMBS and Delayed-Q is much more clear in this case than for the 4x4 grid.
>
> For the M dependency we will create a graph that shows how average reward increases as M grows. In very practical applications there is always a tradeoff between accuracy and speed of computations. This prevents M from being very large.
>
> The Roadrunner game is pretty special. This game has two very different phases. In the first phase the roadrunner moves up and down on a 5 lane highway to avoid incoming traffic. Both SMBS and Delayed-Q are pretty good at learning phase 1. In fact, they learn to bait the coyote into incoming traffic. Phase 2 is very different. It involves jumping over obstacles with no incoming traffic. Both models fail to learn how to deal with jumps as they have already learned a different strategy. Rewards are quite similar, with difference in the values coming from the phase 2 performance. There is really no statistically significant difference in average performance between the two models. Furthermore, in phase 1 the highway is quite long and incoming traffic may be seen way in advance, thus negating the delay penalty. So, the reward is quite comparable despite the increased number of delay steps.
>
> Q2. Indeed, when increasing $\alpha$ the average reward drops. The range 0 to 1 we have in the paper is not enough to show this. We recreated the plot with a range of 0 to 4 and we can clearly see the decay.
>
> Q3. For the discrete state space this is correct. We can estimate the transition probability matrix and estimate the expected Q-value directly. However, in continuous state spaces this is impossible. The step 8 is needed for this case. We are using neural networks to model and estimate the transition function. The algorithm written is the more generic version. We will add a note in the final version to specify how the step are be simplified in the discrete case.

---

### Official Review · Reviewer_aGhW · 2024-01-26

**Significance And Importance:** 2
**Soundness:** 1
**Novelty:** 2
**Clarity:** 3
**Overall Evaluation:** 1
**Confidence:** 3

**Weaknesses:**

1: Minor weaknesses that are easily fixable.

**Contributions Of The Paper:**

The paper provides a method for control in stochastic systems with observation and action delay. The paper addresses the exponential growth of possible states using sampling, and plans for the average predicted state.

**Ethical Considerations:**

(1) Not Applicable: The paper does not have any ethical considerations to address

**Nomination For Best Paper:**

No

**Questions For Authors:**

1. Given that a non-delayed optimal policy exists, and the system is evolving according to the non-delayed optimal policy, what is the motivation for control using a delayed policy?
2. What makes SMBS not a "deterministic" method? Is it simply that it is sampling at each point, and so there exists a distribution over the actual action taken? In that case, would SMBS be a deterministic method in the limit, as the number of samples approach infinity?
3. What were the parameters for running the experiments? Specifically, the experiments note that 5 models were trained for each method. How were the models trained? Were they trained until convergence, or up to a time limit? Why were the results reported for only the top 4 models for each method?

===
Post rebuttal
===
My main concerns were:
- The motivation for finding a delayed control policy, given that the system is evolving already according to actions derived from q*, a "non-delayed optimal Q-function" lines 135-138. This is passed into Algorithm 1 (SMBS Policy) in Line 1, and used in Line 8 of Algorithm 1 to simulate forward. Reading Line 8 of Algorithm 1, the simulation samples state s_{t-d+j+1} from applying a_{t-d+j} at state s_{t-d+j}, with a_{t-d+j} presumably from q*. What is actually trained in Algorithm 2, and used as input into Algorithm 1 is a delayed policy. This should be clarified in the final text text.
- The text states in Line 176 that the authors "term this planning method [Delayed-Q] as deterministic". It was unclear how the proposed method contrasts. It was important to clarify that stochasticity comes from the sampling.

Further minor errors:
- Algorithm 1, line 10: should be updating \mathcal{D} to be \mathcal{D}\cup \left\{s_t^{(i)}\right\}
- Algorithm 1, line 16: should be updating \mathcal{L} to be \mathcal{L}\cup q^*(\left\{s_t^{(i)}\right\},a)

**Reproducibility:**

3: Authors describe the implementation and domains in sufficient detail.

**Strengths Of The Paper:**

- The paper reads well.
- The paper addresses an important and interesting problem.

**Weaknesses Of The Paper:**

- The paper must clarify the policy governing the evolution of the system during training of the delayed policy. The paper originally stated that the evolution of the system state was simulated based on actions from a non-delayed optimal policy, and calculates the delayed policy for controls based on the system evolving according to the optimal policy. If the system is already being controlled according to a non-delayed optimal policy, it is difficult to see why we would want to override that with a delayed control. However, careful examination of the pseudocode shows that the policy used to generate the action sequences are in fact the delayed policy being trained.
- The paper's results could be better explained. AMDP is described as the full problem, whereas SMBS is a sampling approximation which deals with the intractability of uncertainty propagation. It is unusual that an approximate algorithm would yield better results, unless there were unreported limits on computation. The computation limits should be clarified and emphasised.

---

> ### Author Rebuttal · Authors · 2024-01-26
>
> Q1: Even if a solution of non-delayed control policy exists, this policy performs poorly when delay in environment is present. For example, a professional video game player is not able to use his/her regular winning strategy when the ping is high. In fact, applying an optimal non-delayed policy to delayed environments is known in literature as the "memoryless" approach [Schuitema et al. 2010]. Such an approach performs much worse than either AMDP or Delayed-Q method [Derman et al. 2021], therefore we excluded it from our comparison.
>
> Q2: The word ``stochastic'' in our paper is only used in two situations: 1. the transition of the environment is stochastic and 2. the planning in our method is stochastic. We do not claim that our policy is stochastic. Indeed, the policy itself is deterministic given sufficiently large planning samples. The name of the method (SMBS) builds on the MBS method in [Walsh et al. 2009]. The original paper uses a deterministic planning method whereas our method incorporates stochastic planning.
>
> Q3: All training details are in the section ``Training/Evaluation Procedure''. For a fair comparison, all models are trained with the same amount of samples (line 307). Chosen parameters are specified on line 325. DRL methods are sensitive to the choice of initial parameters. Some initial parameter choices lead to a much longer training time until convergence. We discard the worst performing model for all methods in order to obtain robust results. In fact, the AMDP method was the most sensitive method to the choice of initial parameters. If we did not discard the worst model, the results of the AMDP method would be even worse than reported.
>
> Additional answers to the pointed weaknesses of out paper. We believe there are fundamental misunderstandings about our approach. First, our method solves the same problem as the Delayed-Q method does, both methods have highly similar training procedure and model components. Therefore, Delayed-Q is a perfect baseline method for our method. Second, the AMDP method indeed defines the full problem. However, as mentioned in the paper, solving the full problem is not easy at all. In fact, given the same conditions (without knowing the reward function and transition probability function), training AMDP models is much slower, and usually results in worse control policy after being trained with the same amount of samples as the other two methods.

---

### Meta-Review · Area_Chair_BQS3 · 2024-02-02

**Recommendation:** Accept (Poster)
**Confidence:** 3

**Metareview:**

The paper describes a method, called Stochastic Model-Based Simulation (SMBS), for decision-making in with delayed feedback. The key idea is to anticipate a different possible future states rather than relying on deterministic estimates.

The reviewers agreed on the value of the work despite some issues with the experiments and the scolarship. We recommend the authors to include the information provided in the rebuttal in the final version of the paper.

**Ethical Considerations:**

(1) Not Applicable: The paper does not have any ethical considerations to address